# Pyramiding QTLs controlling tolerance against drought, salinity, and submergence in rice through marker assisted breeding

**Valarmathi Muthu, Ragavendran Abbai, Jagadeeshselvam Nallathambi, Hifzur Rahman, Sasikala Ramasamy, Rohit Kambale, Thiyagarajan Thulasinathan, Bharathi Ayyenar, Raveendran Muthurajan**[ORCID]*

Centre for Plant Molecular Biology and Biotechnology Tamil Nadu Agricultural University, Coimbatore, India

* raveendrantnau@gmail.com

**Data Availability Statement:** All relevant data are within the manuscript and its Supporting Information files.

## Abstract

Increases in rice productivity are significantly hampered because of the increase in the occurrence of abiotic stresses, including drought, salinity, and submergence. Developing a rice variety with inherent tolerance against these major abiotic stresses will help achieve a sustained increase in rice production under unfavorable conditions. The present study was conducted to develop abiotic stress-tolerant rice genotypes in the genetic background of the popular rice variety Improved White Ponni (IWP) by introgressing major effect quantitative trait loci (QTLs) conferring tolerance against drought ($qDTY_{1.1}$, $qDTY_{2.1}$), salinity (Saltol), and submergence (Sub1) through a marker assisted backcross breeding approach. Genotyping of early generation backcrossed inbred lines (BILs) resulted in the identification of three progenies, 3-11-9-2, 3-11-11-1, and 3-11-11-2, possessing all four target QTLs and maximum recovery of the recurrent parent genome (88.46%). BILs exhibited consistent agronomic and grain quality characters compared to those of IWP and enhanced performance against dehydration, salinity, and submergence stress compared with the recurrent parent IWP. BILs exhibited enhanced tolerance against salinity during germination and increased shoot length, root length, and vigor index compared to those of IWP. All three BILs exhibited reduced symptoms of injury because of salinity (NaCl) and dehydration (PEG) than did IWP. At 12 days of submergence stress, BILs exhibited enhanced survival and greater recovery, whereas IWP failed completely. BILs were found to exhibit on par grain and cooking quality characteristics with their parents. Results of this study clearly demonstrated the effects of the target QTLs in reducing damage caused by drought, salinity, and submergence and lead to the development of a triple stress tolerant version of IWP.

## Introduction

The global population is expected to reach nine billion by 2050 from the current total of seven billion, and this will necessitate increased food production by at least 50%. Total production of the major cereal food crop, rice, has to reach 160 million tons by 2050 from the current level of

**Funding:** Corresponding author RM received funding support from Department of Biotechnology, Government of India, New Delhi (Grant # BT/PR13454/COE/34/43/2015).

**Competing interests:** The authors have declared that no competing interests exist.

100 million tons [1]. Recent trends in the productivity of major food crops suggest that current growth in productivity will not be sufficient to meet the predicted demand [2]. Exacerbating the problem is the decline in natural resources and climate change, both of which pose serious threats to a sustained increase in agricultural productivity [3]. A sustained increase in rice productivity necessitates the development of resilient rice varieties adapted to marginal environments where the yield gap is high because of various biotic/abiotic stresses. Even though rice is a tropical crop, it exhibits an extreme susceptibility to major abiotic stresses *viz.*, drought, salinity, and submergence [4]. Drought affects nearly 42 Mha of rice grown under rainfed lowlands and uplands leading to a yield loss of 13–35% every year [5]. Salinity ranks second among the major abiotic stresses affecting rice growth and productivity in coastal and marginal inland environments [6]. Even very mild to moderate salinity (EC = 5–6 dS/m) can cause significant yield losses in rice [7]. In India, salinity affects rice cultivation in ~20% of irrigated lands and causing yield loss of up to 45% [8]. Rice, a water loving crop, can tolerate brief periods of submergence; however, prolonged submergence beyond 8–9 days will have significant effects on rice establishment and yield. Because of the changing climate, flooding/submergence often poses a serious threat to rice cultivation in rainfed lowlands, which occupy 25 Mha of rice cultivation area in South and Southeast Asia [9]. In India, approximately 12–14 Mha (30%) of rice growing area is frequently affected by submergence/flash flooding where the average productivity ranges from 0.5–0.8 t ha$^{-1}$.

The majority of the popular rice varieties are susceptible to the above three abiotic stresses and widen the yield gap between potential and realized yield under marginal environments. Genetic improvement of rice for abiotic stress tolerance through conventional approaches has met with limited success because of the complex nature of mechanisms governing these traits and problems associated with phenotyping the traits involved. Recent advancements in molecular genetics and genotyping led to the identification of major effect quantitative trait loci (QTLs) linked to major abiotic stresses viz., drought tolerance ($qDTY_{1.1}$, $qDTY_{2.1}$, $qDTY_{3.1}$ and $qDTY_{6.1}$ in Apo [10–12] and $qDTY_{12.1}$ in Way rarem [13], submergence tolerance *Sub1* in FR13A [14], and salinity tolerance *Saltol* in Pokkali [15]. Deployment of the above QTLs through molecular breeding paved the way for the development and release of a submergence-tolerant version of a popular rice variety Swarna [16], submergence-tolerant version of CO 43 [17], drought-tolerant version of IR64, namely DRR Dhan 42, drought-tolerant MR219 [5], drought-tolerant Sabitri [18], salinity-tolerant version of BR28 [19], salinity-tolerant Pusa Basmati 1 [20], and salinity-tolerant Improved White Ponni (IWP) [21] through the marker assisted back cross breeding (MABB) approach. However, very little research has been conducted to develop multiple abiotic stress-tolerant rice genotype(s) by pyramiding QTLs that control tolerance against drought, salinity, and submergence through marker assisted selection (MAS). In this study, we attempted to develop early generation back cross inbred lines (BILs) of a popular rice variety Improved White Ponni pyramided with QTLs that control tolerance against drought ($qDTY_{1.1}$ and $qDTY_{2.1}$), salinity (*Saltol*), and submergence (*Sub1*) stress. Developed BILs were tested for enhanced tolerance against drought, salinity, and submergence relative to their recurrent parent, IWP.

## Materials and methods

### Genetic materials used

A popular, high yielding, medium-duration rice variety namely Improved White Ponni was selected as the recurrent parent [22]. IWP is a popular rice variety among farmers in South India because of its high yield, fine grain, and high cooking quality; however, it is highly

**Table 1. Details on Target QTLs, traits and their respective donors.**

| S. No | Target Trait | Target QTL | Target QTL | Reference |
|---|---|---|---|---|
| 1 | Drought tolerance | $qDTY_{1.1} qDTY_{2.1}$ | Apo | [10, 11] |
| 2 | Salinity tolerance | Saltol | Pokkali FL478 | [15, 23] |
| 3 | Submergence tolerance | Sub1 | FR13A | [14, 24] |

susceptible to most of the abiotic/biotic stresses, including drought, salinity, and submergence. Target QTLs for the above traits and their respective donors are provided in Table 1.

An already developed back cross progeny of IWP (possessing >90% of the IWP genome), namely # 36-27-2, harboring QTLs controlling tolerance against salinity (Saltol from FL478) and submergence (Sub1 from FR13A) was selected as one parent and an advanced recombinant inbred line (RIL) derived between IWP × Apo, namely CBMAS14065, harboring two major drought-tolerant QTLs ($qDTY_{1.1}$ and $qDTY_{2.1}$) of Apo was selected the other another parent. The $F_{1s}$ derived through inter-mating of the above two lines were back crossed with IWP to develop BILs of IWP exhibiting tolerance against the three major abiotic stresses (Fig 1). Retention of four different target QTLs *viz.*, $qDTY_{1.1}$, $qDTY_{2.1}$, Saltol, and Sub1, was monitored using simple sequence repeats (SSR) markers.

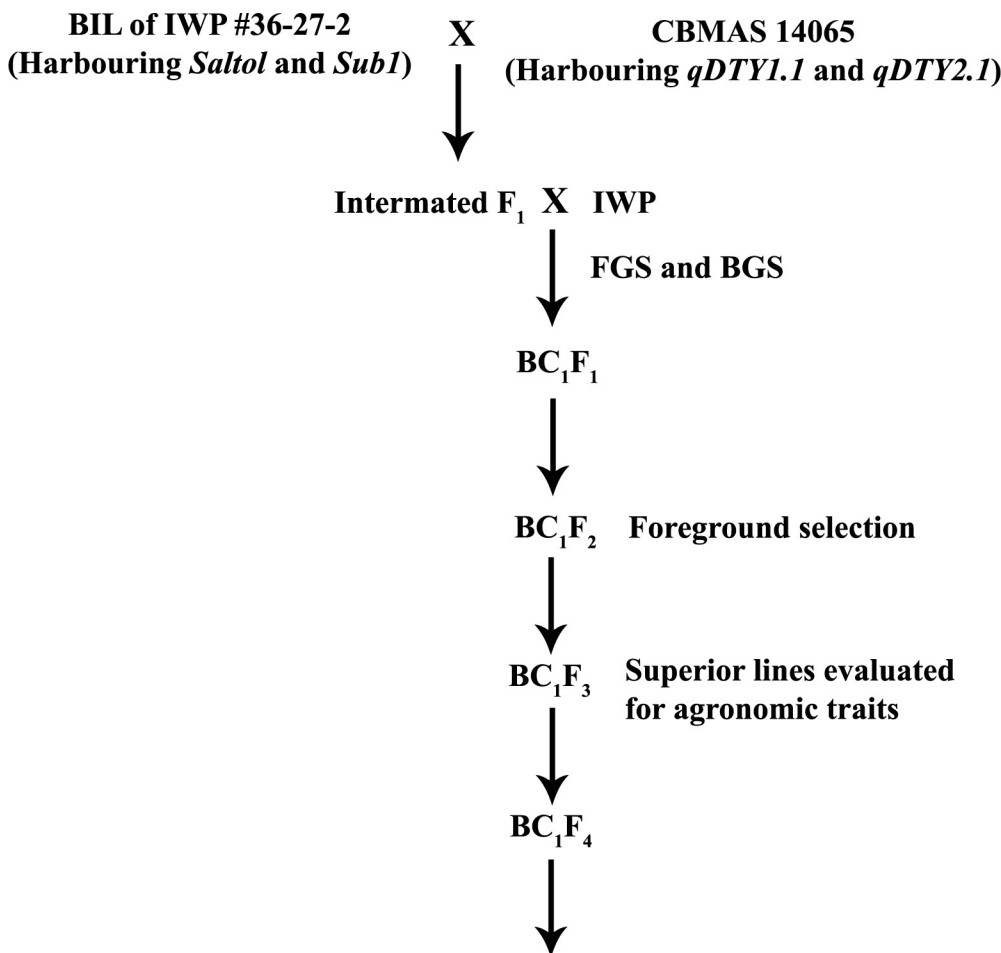

**Fig 1. Pseudo back cross breeding plan for the development of multiple abiotic stress-tolerant version of IWP.**

## Development of BILs of IWP harboring abiotic stress tolerant QTLs

**Parental polymorphism survey.** Thirty-eight SSR markers located in the vicinity of the target QTLs viz., $qDTY_{1.1}$, $qDTY_{2.1}$, *Saltol*, and *Sub1*, were surveyed for polymorphism between the recurrent parent IWP and respective donors for the identification of polymorphic SSRs to be used in foreground selection (FGS). A total of 347 SSR markers, covering all 12 chromosomes, including the carrier chromosomes 1, 2, and 9 were surveyed for polymorphism to identify SSR markers for background selection (BGS).

**Genotyping procedure.** Genomic DNA was extracted from the fresh leaf tissues of all the back cross progenies and parents using the modified CTAB protocol [25]. The quality of DNA was checked by agarose gel electrophoresis and quantified using a Nanodrop ND-1000 spectrophotometer (Thermo Fisher Scientific, Wilmington, DE, USA). PCR amplification was performed in 15 μl reactions containing 25–50 ng of DNA template, 8.0 μl of sterile water, 1.5 μl of assay buffer (10 X), 0.50 μl of 2.5 mM dNTPs, 0.20 μl (3 U/μl) of *Taq* DNA polymerase, and 1.00 μl each of the 10 μM forward primer and reverse primer (initial denaturation at 94˚C for 5 min followed by 36 cycles of 94˚C for 1 min, annealing at 55–58˚C for 1 min, extension at 72˚C for 1 min, followed by a final extension at 72˚C for 10 min). PCR products were resolved by using 3% agarose gel electrophoresis in 1X TBE buffer and bands were visualized after ethidium bromide staining and documented using a gel documentation system (BIO-RAD, USA) for scoring the allelic pattern.

## Generation of BILs of IWP pyramided with QTLs related to drought, salinity and submergence tolerance

The $F_1$s plants developed between CBMAS14065 and BIL # 36-27-2 were confirmed using polymorphic SSR markers and used for further back crossing. All the $BC_1F_1$ plants were subjected to FGS using SSR/InDel markers listed in Table 2 and positive $BC_1F_1$ plants were subjected to BGS using 42 polymorphic SSR markers covering all 12 chromosomes. The $BC_1F_1$ plants with the maximum recovery of the recurrent parent (IWP) genome were selected and advanced to the $BC_1F_2$ generation. All the $BC_1F_2$ progenies were subjected to FGS and progenies found to be homozygous for all 4 target loci were forwarded to the $BC_1F_3$ generation and evaluated for agronomic performance under field conditions. Superior single plants were selected from each of the $BC_1F_3$ families and used to study responses against drought, salinity, and submergence stresses.

**Evaluation of BILs against salinity stress.** Thirty seeds in each of the BILs viz., BIL # 3-11-9-2, BIL # 3-11-11-1, and BIL # 3-11-11-2 were germinated along with IWP and FL478 in Petri plates containing 10 ml of varying concentrations of NaCl solution (80 mM, 100 mM, and 125 mM) by maintaining suitable controls. Germination % for all the genotypes was recorded on the 5th day after sowing. On the 9th day, lengths of shoots and roots were measured from 10 randomly selected seedlings and used to calculate the vigor index [26]. Similarly, all three BILs were evaluated for their vegetative stage responses against salinity stress using the modified standard evaluation scoring (SES) system of IRRI [27]. BILs were germinated in

**Table 2. Details of markers and their primer sequences used for foreground selection.**

| Markers | Primer Sequence | Annealing temperature ˚C |
|---------|-----------------|--------------------------|
| RM 472 | CCATGGCCTGAGAGAGAGAG (F) AGCTAAATGGCCATACGGTG (R) | 55 |
| RM 2634 | GATTGAAAATTAGAGTTTGCAC (F) TGCCGAGATTTAGTCAACTA (R) | 55 |
| RM 3412 | AAAGCAGGTTTTCCTCCTCC (F) CCCATGTGCAATGTGTCTTC (R) | 55 |
| ART 5 | CAGGGAAAGAGATGGTGGA (F) TTGGCCCTAGGTTGTTTCAG (R) | 58 |

Petri dishes along with their parents and 7-day-old seedlings were then transferred to hydroponic cultures (Yoshida solution) for establishment. Twenty-one day old seedlings were subjected to salinity stress by adding NaCl into the Yoshida solution at the final concentration of 100 mM NaCl. Responses of BILs against salinity stress were recorded based on leaf rolling and drying symptoms in comparison with the recurrent parent IWP. To quantify the amount of salts ($Na^+$ and $K^+$) accumulated in shoot tissues of IWP, FL 478, and the 3 BILs, 0.5 g of dried shoot tissues were digested using a tri-acid mixture ($HNO_3$: $H_2SO_4$: $HClO_4$ in the ratio of 9:2:1) in a digestion block at 200°C for 3 h. The digested mixture was filtered and $Na^+$ and $K^+$ contents were measured using a Labtronics Microprocessor Flame Photometer LT-671 [28]

**Evaluation of BILs against dehydration stress.** Seeds of IWP, FL478, and the 3 BILs were surface sterilized using sodium hypochlorite and germinated in Petri plates. Germinated seedlings were transferred to hydroponic conditions (Yoshida solution) and grown up to 21 days. Seedlings were subjected to dehydration stress simulated by -0.5MPa polyethylene glycol (PEG 6000). After 21 days, plants were scored for their responses against osmotic stress using the Standard Evaluation System [29].

**Evaluation of BILs against submergence.** Seedlings of IWP and the 3 BILs were raised in plastic pots (17 cm height and 16 cm diameter) filled with a field soil mixed with the required amount of nutrients, 1.25 g of $(NH_4)_2SO4$, 0.08 g muriate of potash (KCl), and 0.08 g single superphosphate (SSP), and plants were maintained free of pests and diseases. Twenty-day-old seedlings were subjected to submergence stress by keeping the pots inside 1.5 m tanks filled with water [17]. Survival of seedlings was assessed by taking out the seedlings after 12 days of submergence and their ability to recover was determined.

**Grain and cooking quality analysis of BILs.** Grains of 3 BILs and their parents were evaluated for quality traits like length and breadth of milled grains, kernel length after cooking (KLAC), kernel breadth after cooking (KBAC), kernel length elongation ratio, kernel breadth elongation ratio [30], volume expansion ratio, gelatinization temperature [31], gel consistency and amylose content [32, 33]. Five fully developed wholesome milled rice kernels were taken in all samples and their length and breadth were measured using digital Vernier calliper. The kernel length and breadth after cooking (KLAC and KBAC) were measured by soaking 25 whole milled kernels in 20ml of water and boiling them in water bath maintained at 98°C for 10 min, the cooked rice was transferred to petridish lined with filter paper and then length and breadth of 10 cooked whole grains were measured [34]. The kernel length and breadth elongation ratio (KER and BER) were expressed as the ratio of the average length and breadth of the cooked kernels to that of the uncooked kernels respectively [34]. The volume expansion of milled and cooked rice was measured by water displacement method in a graduated measuring cylinder. The volume expansion ratio (VER) was expressed as the ratio of the volume of cooked rice to that of uncooked milled rice [35]. The gel consistency of of BILs along with parents was measured as described by Cagampang et al. [33]. Hundred milligram of rice powder with 12% moisture was placed in 13 x 100mm culture tubes and wetted with 95% ethanol containing 0.025% thymol blue and 2ml of 0.2N KOH was added to the samples. The tubes containing samples were heated in boiling water bath for 10 min and then cooled in ice water bath for 20 min. Gel consistency was measured by the length of cold gel in test tubes held horizontally on graph paper after 30 min. The gelatinization temperature was obtained by measuring the alkali spreading value. Five whole milled kernels were incubated in 10 ml of 1.7% KOH solution in a Petri dish for 24 h at 30°C and scored on a 1–7 scale as described by Little [31]. Amylose content of rice flour of BILs and their respective parents was determined following the modified method of Juliano [36]. One milliliter ethanol (95%) and 9 mL of sodium hydroxide (1N) were added to 0.1 g of rice flour. The samples were heated on a boiling water bath

**Table 3. Details of markers identified for foreground selection of target QTLs.**

| Donor | Trait | Target QTLs and its position (Mb) | Chromosome | Markers used for FGS and its position (Mb) |
|---|---|---|---|---|
| APO | Drought | $qDTY_{1.1}$ (34.9–37.8) | 1 | RM 472 (37.8) |
| | | $qDTY_{2.1}$ (11.4–20.7) | 2 | RM 2634 (20.4) |
| FL478 | Salinity | *Saltol* (10.7–12.2) | 1 | RM 3412 (11.5) |
| FR13A | Submergence | *Sub1* (4.5–7.2) | 9 | ART 5 (6.4) |

followed by cooling for 1 h, distilled water was added to make the final volume of 100 ml. One millilitre of acetic acid (1 N) and 2 ml of freshly prepared iodine solution were added to 5 mL of the stock sample solution. Absorbance of the solution was measured at 620 nm after 20 min.

## Results

### Parental polymorphism survey

Survey of parental polymorphisms using 38 SSR markers located in the vicinity of the four target QTLs resulted in the identification of 4 markers for FGS (Table 3).

Parental polymorphism survey using 347 genome wide polymorphic SSR markers resulted in the identification of 58 polymorphic SSRs covering all 12 chromosomes for BGS (Fig 2).

### Development of BILs of IWP pyramided with QTLs controlling tolerance against drought, salinity, and submergence

CBMAS14065, a RIL of IWP × Apo harboring two major effect drought tolerant QTLs $qDTY_{1.1}$ and $qDTY_{2.1}$ was crossed with a NIL of IWP, namely # 36-27-2 harboring *Saltol* from FL478 and *Sub1* from FR13A. Genotyping of 94 $F_1$ progenies using RM472 ($qDTY_{1.1}$), RM2634 ($qDTY_{2.1}$), RM3412 (*Saltol*), and ART5 (*Sub1*) identified 5 true $F_1$s plants, which were back-crossed with IWP to produce 87 $BC_1F_1$ plants. FGS of $BC_1F_1$s resulted in the identification of 4 $BC_1F_1$ progenies harboring all four QTLs *viz.*, $qDTY_{1.1}$, $qDTY_{2.1}$, *Saltol*, and *Sub1* (Fig 3). BGS of four $BC_1F_1$ plants using 42 genome-wide SSR markers revealed that the recovery of the recurrent parent genome ranged from 76.10 to 88.46%. The $BC_1F_1$ progeny # 3–11 retaining 88.46% of the IWP genome was selfed and used to raise $BC_1F_2$ (Fig 4). FGS of 90 $BC_1F_2$ progenies resulted in the identification of 2 plants harboring all four QTLs under homozygous condition *viz.*, 3-11-9 and 3-11-11. Evaluation of 129 $BC_1F_3$ progenies of the above two

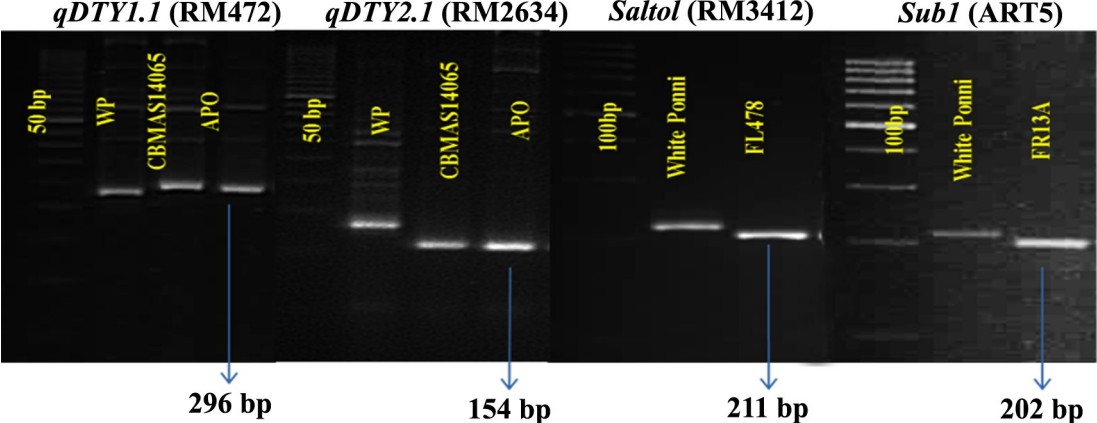

**Fig 2. Parental polymorphism for the markers linked to the target QTLs.**

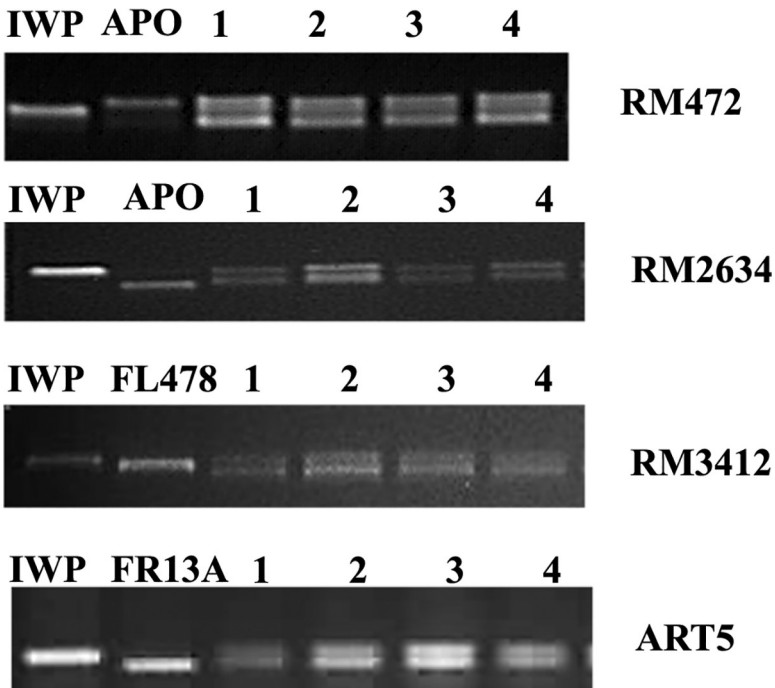

**Fig 3. Foreground selection of $BC_1F_1$ progenies using markers linked to target traits.**

lines resulted in the identification of 3 superior single plant progenies viz., 3-11-9-2, 3-11-11-1, and 3-11-11-2 (Table 4). BGS using 58 polymorphic SSR markers revealed the recovery of 89.65% of the IWP genome in all three progeny.

## Evaluation of BILs for their performance under field condition

BILs were evaluated for their agronomic performance along with IWP over two seasons (Kharif 2017 and Kharif 2018) under field conditions. All three BILs were found to be consistent with that of IWP in plant height, the number of tillers, days to flowering, grain yield per plant, and 100 grain weight (Table 4). BILs (94.2 cm to 100.7 cm) were 30 cm shorter than IWP (133.6 cm).

**BILs exhibited increased germination under salinity.** Three BILs harboring *Saltol* loci were evaluated for their ability to germinate under salinity. At 100 mM NaCl stress, only 40% of susceptible IWP germinated, whereas all the 3 BILs exhibited a significantly higher germination rate (Table 5; Fig 5). FL478 exhibited a maximum germination of 86.7% at 100 mM concentration followed by 3-11-11-1 (83.3%), 3-11-11-2 (76.7%), and 3-11-9-2 (73.3%). At 125 mM NaCl stress, BIL # 3-11-11-1 exhibited greater germination at 80% and was consistent with that of the donor parent (Table 5; Fig 5). Salinity stress had significant effects on the growth of seedlings in all genotypes, but the effects were more severe in IWP (Fig 5). All 3 BILs exhibited significantly higher seedling vigor over relative to their recurrent parent IWP under varying levels of salt stress (Table 5).

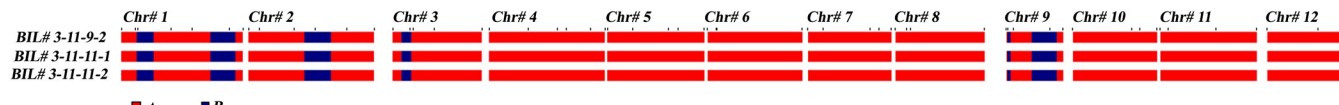

**Fig 4. Graphical genotyping of 3 superior BILs showing the extent of IWP genome recovery.**

**Table 4. Agronomic performance of BILs under field conditions.**

| Entries | Plant height (cm) | Number of tillers | Days to flowering | Yield per plant (g) | 100 grain weight (g) |
|---|---|---|---|---|---|
| IWP | 133.6 ± 2.5 | 23.4 ± 1.0 | 108.7 ± 0.9 | 27.94 ± 0.3 | 1.69 ± 0.1 |
| BIL # 3-11-11-1 | 99.3 ± 1.4 | 25.6 ± 0.9 | 102.7 ± 1.2 | 29.76 ± 0.9 | 1.65 ± 0.2 |
| BIL # 3-11-11-2 | 100.7 ± 1.7 | 24.8 ± 0.7 | 103.7 ± 1.2 | 31.71 ± 0.3 | 1.76 ± 0.1 |
| BIL # 3-11-9-2 | 94.2 ± 2.2 | 23.6 ± 0.5 | 108.3 ± 0.9 | 30.88 ± 0.8 | 1.75 ± 0.1 |

**BILs exhibited enhanced tolerance against salinity under hydroponic conditions.** Seedlings of 3 BILs harboring the *Saltol* loci from FL478 exhibited enhanced tolerance against 100 mM NaCl stress as compared to that of IWP under hydroponics conditions. IWP exhibited extreme susceptibility to 100 mM NaCl stress with an average SES score of 9, whereas the tolerant FL478 displayed an average SES score of 3. All three BILs exhibited moderate tolerance against 100 mM NaCl stress and exhibited an average SES score of 5 (Fig 6). Genotypes did not differ in their shoot $K^+$ content but differed in their shoot $Na^+$ content. The amount of $K^+$ in the shoot tissues of IWP was lower than that of FL478 and the 3 BILs (Table 6). The $Na^+/K^+$ ratio under salinity stress was highest in the susceptible IWP (3.21) and lowest in the donor FL478 (1.45). BILs were intermediate between the parents in the maintenance of the $Na^+/K^+$ ratio and it ranged from 1.95–2.29 (Table 6).

**BILs exhibited enhanced tolerance against dehydration relative to that of IWP** BILs exhibited enhanced tolerance against dehydration (imposed by -0.5 MPa PEG under hydroponic conditions) when compared to that of IWP. IWP exhibited extreme susceptibility score of 9, whereas all 3 BILs performed better than did IWP with an SES score of 5 (Fig 7; Table 7). All the genotypes exhibited consistent results in the shoot ratio under control conditions and PEG treatment, with significant effects on the shoot and root lengths. After 28 days of stress, the root/shoot ratio in the BILs ranged from 0.56–0.58, whereas IWP produced the lowest ratio of 0.42 (Fig 8).

**BILs exhibited enhanced tolerance against submergence relative to that of IWP.** Seedlings of BILs harboring the *Sub1* loci exhibited enhanced levels of tolerance against 12 days of submergence over that of their recurrent parent IWP (Fig 9). Seedlings of IWP showed 0% survival upon de-submergence after 12 days of stress. Leaves of donor FR13A and BILs remained green as compared to that of IWP and exhibited 66–80% recovery upon de-submergence (Table 8).

**Grain quality traits of BILs.** Grain quality parameters (before and after cooking) of the BILs were evaluated against the recurrent parent (Fig 10; Table 9). BILs were found to possess medium slender grains similar to their recurrent parent. The KLBC of the BILs ranged from 5.0 mm (BIL 3-11-11-1) to 5.1 mm (BIL 3-11-9-2) as compared to 5.4 mm in IWP and 5.2 mm

**Table 5. Effects of salt stress on seed germination of parents and BILs.**

| Entries | Control | | | 80 mM | | | 100 mM | | | 125 mM | | |
|---|---|---|---|---|---|---|---|---|---|---|---|---|
| | GP (%) | SL (cm) | SVI | GP (cm) | SL (cm) | SVI | GP% | SL (cm) | SVI | GP (%) | SL | SVI |
| IWP | 96.7 | 6.6 ± 0.1 | 6.4 | 66.7 | 1.2 ± 0.0 | 0.8 | 40 | 0.9 ± 0.1 | 0.4 | 26.7 | 0.6 ± 0.0 | 0.2 |
| FL478 | 100 | 11.6 ± 0.4 | 11.6 | 93.3 | 6.2 ± 0.2 | 5.8 | 86.7 | 3.5 ± 0.1 | 3.0 | 86.7 | 2.7 ± 0.0 | 2.3 |
| BIL # 3-11-9-2 | 93.3 | 8.3 ± 0.3 | 7.7 | 80 | 3.5 ± 0.2 | 2.8 | 73.3 | 2.9 ± 0.1 | 2.1 | 66.7 | 1.7 ± 0.0 | 1.1 |
| BIL # 3-11-11-1 | 96.7 | 8.2 ± 0.1 | 7.9 | 93.3 | 4.1 ± 0.2 | 3.8 | 83.3 | 3.4 ± 0.1 | 2.8 | 80 | 2 ± 0.1 | 1.6 |
| BIL # 3-11-11-2 | 100 | 7.7 ± 0.1 | 7.7 | 96.7 | 3.7 ± 0.0 | 3.6 | 76.7 | 3.1 ± 0.1 | 2.4 | 73.3 | 1.8 ± 0.0 | 1.3 |

GP, Germination percentage; SL, Length of seedlings; SVI, Seed Vigor Index

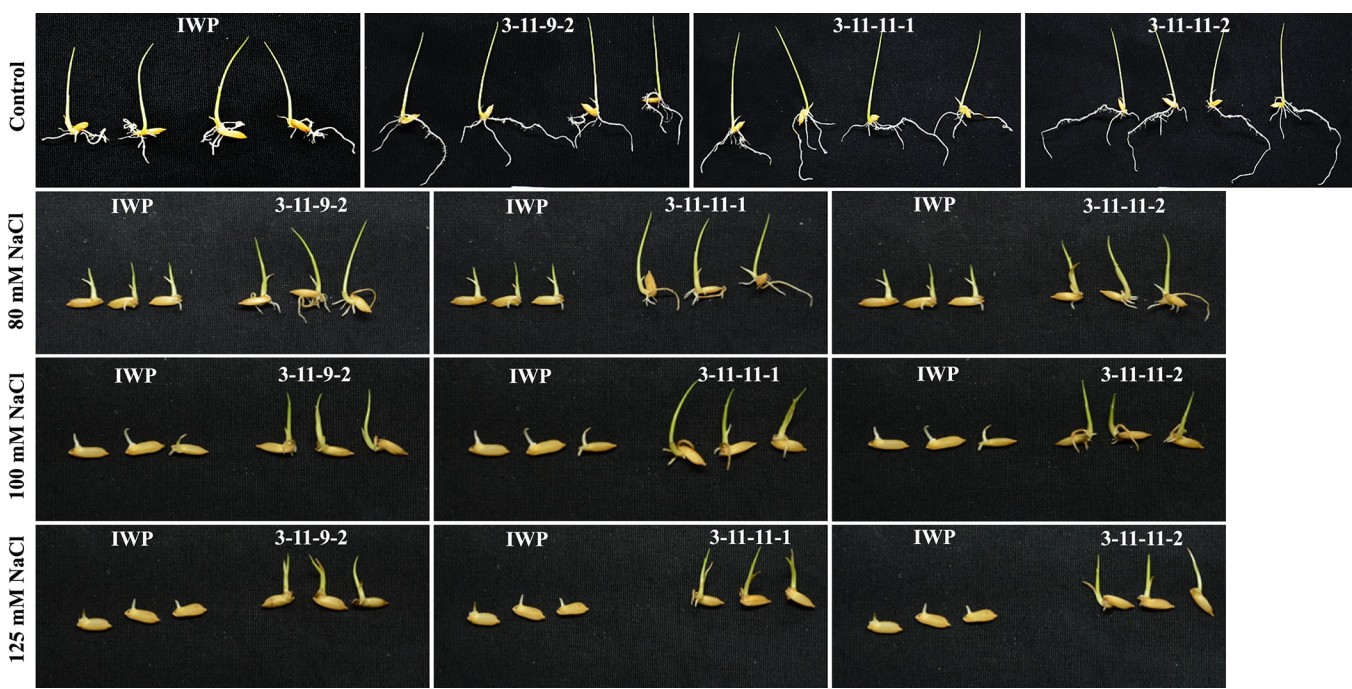

**Fig 5. Performance of BILs and IWP under different levels of salinity during germination.**

in CBMAS14065; whereas the KLBR of BILs ranged between 1.7–1.8 as compared to CBMAS14065 (1.8) and IWP (2.0) respectively. Two BILs had superior KLERC i.e. 1.88 (BIL 3-11-9-2) and 1.92 (BIL 3-11-11-1) as compared to IWP (1.78) and CBMAS14065 (1.85). The gelatinization temperature was found to be intermediate (69–74°C) in the BILs as that of their parents; whereas gel consistency was found to be soft for BILs (90-100mm) and was found to be on par with the parents IWP and CBMAS14065. The amylose content of BILs ranged from 23–24.5% whereas it was found to be 23 and 24% for CBMAS14065 and IWP respectively.

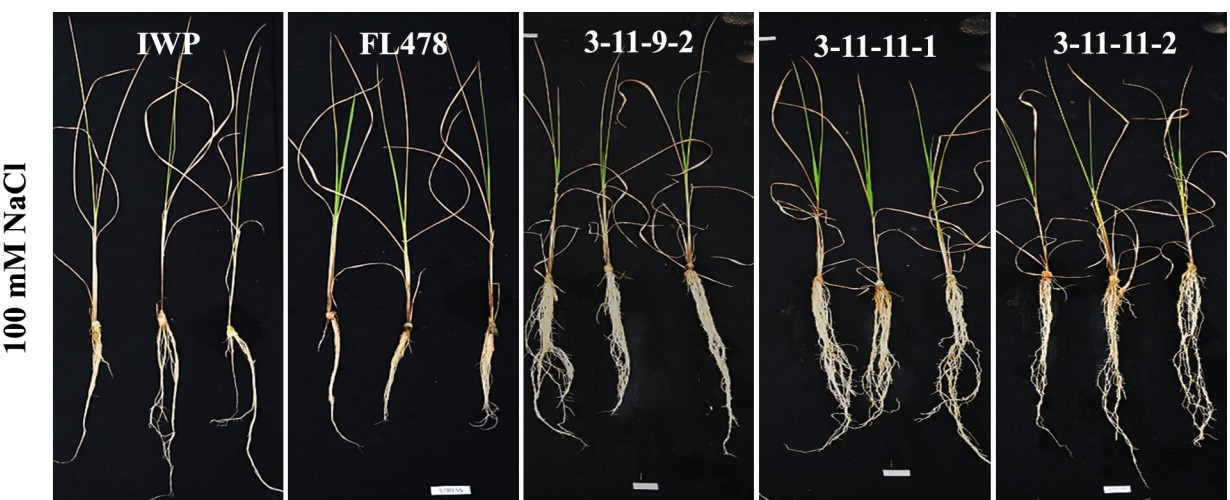

**Fig 6. Performance of BILs and IWP against 100 mM NaCl stress under hydroponic conditions.**

**Table 6. Shoot Na⁺ and K⁺ concentration (mg/g) and Na⁺/K⁺ ratios of BILs and parents under control and salt stress (100 mM) conditions using hydroponic solution.**

| Entries | Control | | | Stress | | | Salt Injury Score |
|---|---|---|---|---|---|---|---|
| | Na | K | Na/K ratio | Na | K | Na/K ratio | |
| IWP | 20.1 ± 0.4 | 51.7 ± 0.4 | 0.38 ± 0.0 | 88.4 ± 0.3 | 27.5 ± 0.6 | 3.21 ± 0.0 | 9 |
| FL478 | 6.2 ± 0.5 | 56.5 ± 0.6 | 0.12 ± 0.0 | 43.1 ± 0.5 | 29.7 ± 0.4 | 1.45 ± 0.0 | 3 |
| BIL # 3-11-9-2 | 7.3 ± 0.3 | 58.4 ± 0.4 | 0.13 ± 0.0 | 65.3 ± 0.3 | 28.4 ± 0.4 | 2.29 ± 0.0 | 5 |
| BIL # 3-11-11-1 | 6.5 ± 0.2 | 59.4 ± 0.5 | 0.12 ± 0.0 | 56.4 ± 0.3 | 28.9 ± 0.8 | 1.95 ± 0.0 | 5 |
| BIL # 3-11-11-2 | 6.6 ± 0.2 | 56.6 ± 0.4 | 0.12 ± 0.0 | 66.3 ± 0.2 | 32.7 ± 0.5 | 2.03 ± 0.0 | 5 |

## Discussion

Global food production must increase by 50% by 2050 to meet food and fuel requirement of the increasing human population [37]. Rice is one of the major cereal food crops whose production has to be doubled to achieve the projected demand [38] and current yield trends are not sufficient to meet the projected growth [2]. Drought, salinity, and flooding are considered the top three major abiotic stresses globally affecting rice production. Among the three stresses, drought and flooding are more often observed within the same growing season, particularly in the rainfed ecosystems of South and Southeast Asia [39]. Heavy downpours during the onset of monsoons may bring flash floods and prolonged dry phases because of the early withdrawal of the monsoons, which may lead to drought during the later stages of rice growth [4]. Apart from drought and flood, salinity is one of the major stresses limiting rice yield, especially in the coastal areas of South and Southeast Asia [40]. Improved rice varieties with high-yield potential and tolerance to drought, salinity, and submergence stresses will allow us to enhance rice productivity in these marginal environments [41]. Physiological and genetic complexity of tolerance mechanisms against the above three abiotic stresses makes conventional breeding a challenging route by which to develop triple tolerant rice [42].

Recent advances in marker technology enabled rapid generation of improved knowledge of the molecular genetic basis of tolerance against submergence, drought, and salinity stresses in rice. Efforts at IRRI, Philippines and UC Davis, USA led to the identification of a major QTL, namely *Sub1*, derived from an Indian landrace FR13A, which confers tolerance against 2–3 weeks of flash floods/submergence [43]. *Sub1* has been extensively deployed in MAB programs

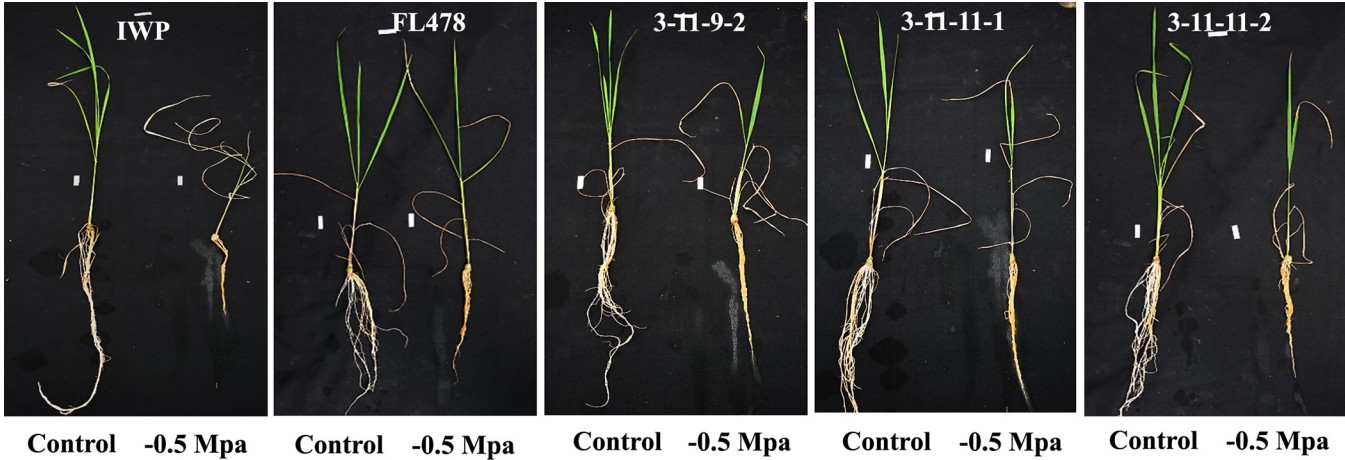

**Fig 7. Screening for drought tolerance using hydroponics.**

**Table 7. Drought sensitivity scores of parents and BILs [29].**

| Entries | QTLs | Score | Leaf rolling | Leaf drying |
|---|---|---|---|---|
| Improved White Ponni | - | 9 | Leaves tightly rolled (V-shape) | Plant completely dried |
| APO | $qDTY_{1.1}$, $qDTY_{1.2}$, $qDTY_{1.3}$, $qDTY_{2.1}$, $qDTY_{3.1}$, and $qDTY_{6.1}$ | 1 | Leaves start to fold (shallow) | Slight tip drying |
| BIL#3-11-9-2 | $qDTY_{1.1}$, $qDTY_{2.1}$, *Saltol*, and *Sub1* | 5 | Leaves fully cupped (U-shape) | One-fourth to ½ of all leaves dried. |
| BIL#3-11-11-1 | $qDTY_{1.1}$, $qDTY_{2.1}$, *Saltol*, and *Sub1* | 5 | Leaves fully cupped (U-shape) | One-fourth to ½ of all leaves dried. |
| BIL#3-11-11-2 | $qDTY_{1.1}$, $qDTY_{2.1}$, *Saltol*, and *Sub1* | 5 | Leaves fully cupped (U-shape) | One-fourth to ½ of all leaves dried. |

to develop submergence-tolerant versions of popular rice varieties [17, 24, 43, 44]. Similarly, efforts towards mapping of drought tolerance traits in rice at IRRI, Philippines identified several major effect QTLs in different mapping populations [11–13, 45]. Out of several QTLs, few, including $qDTY_{1.1}$ and $qDTY_{2.1}$, from the drought tolerant genotype, Apo, showed consistent effects across different genetic backgrounds and has been used in breeding applications. Similar to the genetic dissection of tolerance against drought and submergence, the QTL mapping approach has been deployed to unravel the genetic complexity of salinity tolerance in rice. Gregorio, Senadhira [46] identified a major QTL *Saltol* in a salinity tolerant Pokkali responsible for maintaining $Na^+/K^+$ homeostasis at the seedling stage. Introgression of the *Saltol* loci from

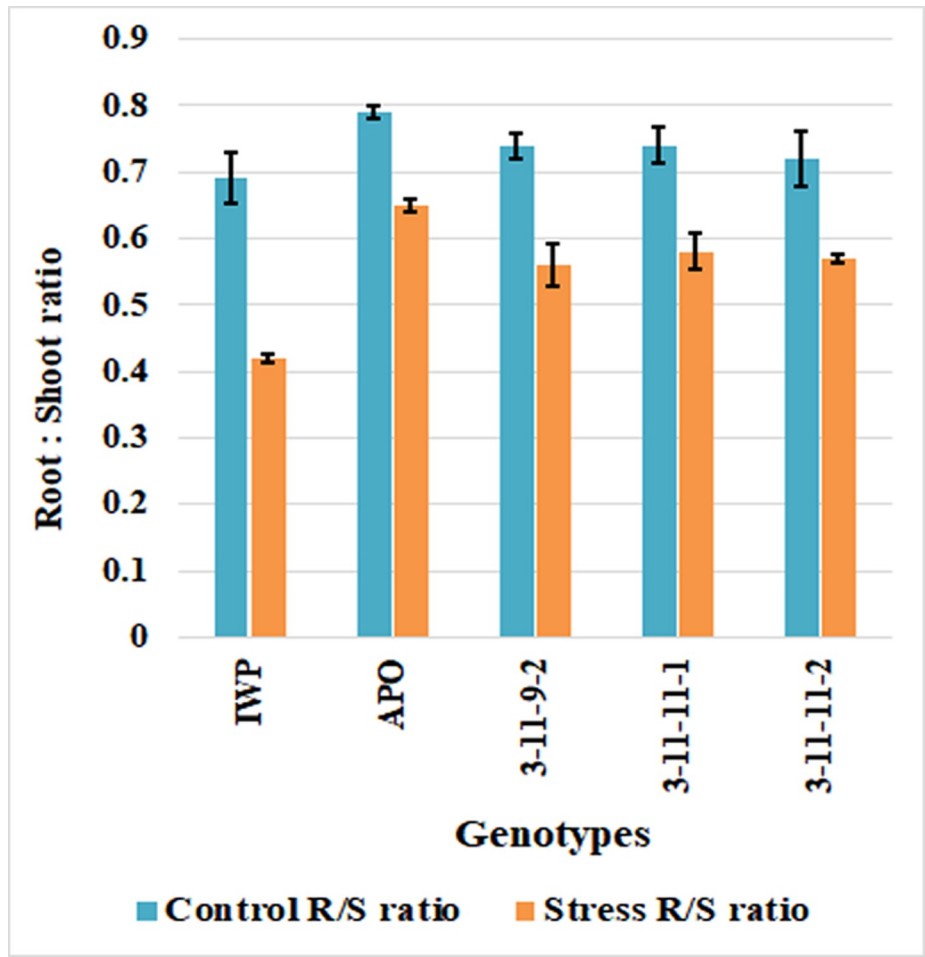

**Fig 8. Differential response of BILs and parents under control and stress (-0.5 MPa PEG) conditions.**

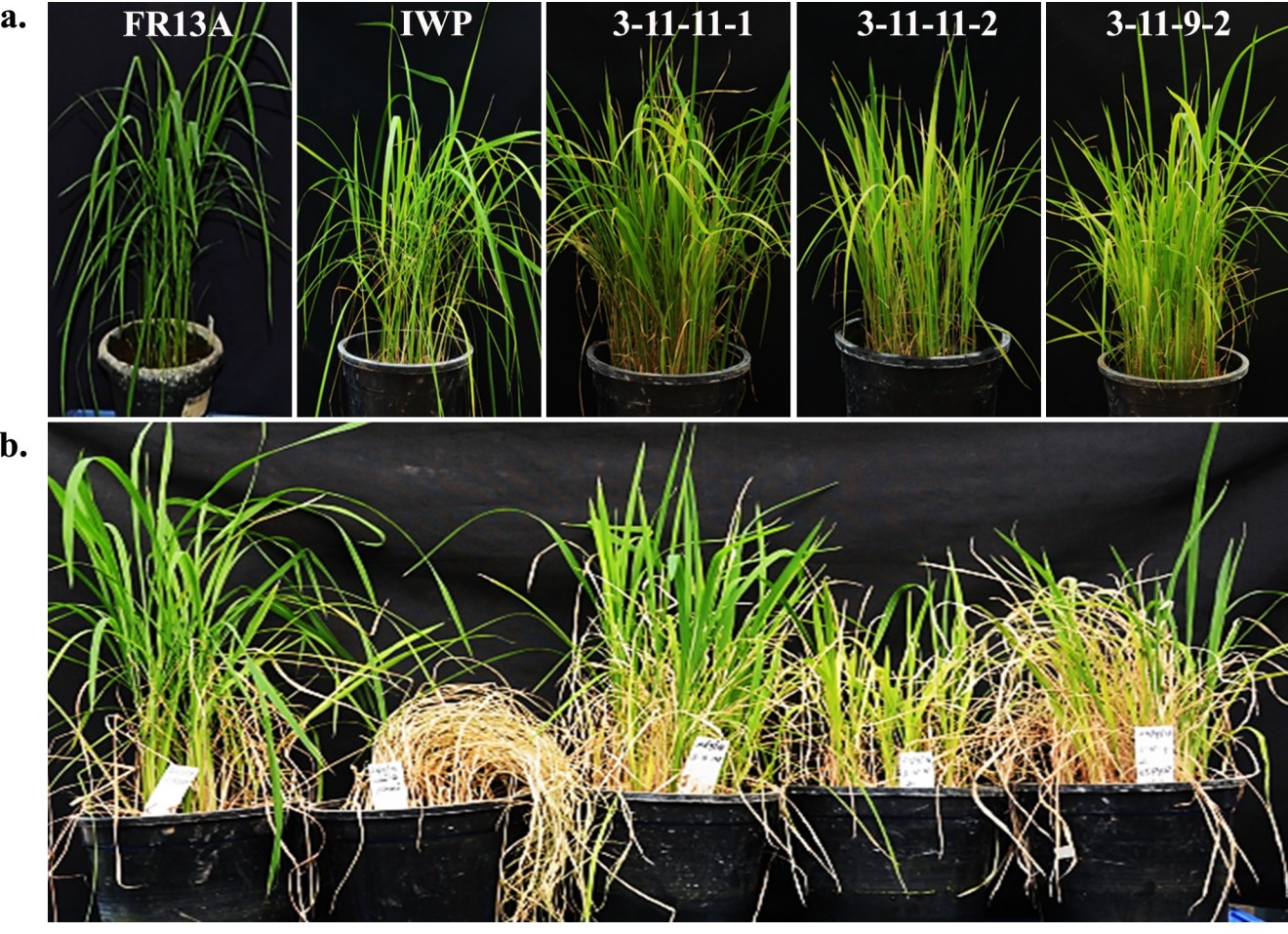

**Fig 9.** Responses of BILs and parents under control conditions (a) and 12 days after de-submergence (b).

Pokkali or its derivative, FL478, through MABB enabled the development of a salinity-tolerant version of popular rice varieties [20, 21].

IWP is one of the popular rice varieties of South India known for its high yield, superior grain quality, and excellent cooking quality. IWP was released during the 1980s and it is still popular among farmers because of the lack of a suitable replacement. However, IWP is highly susceptible to major biotic/abiotic stresses and sustaining IWP cultivation requires genetic enhancement of its stress tolerance. In the present study, a MAB approach was adapted to pyramid major effect QTLs controlling grain yield under drought ($qDTY_{1.1}$ and $qDTY_{2.1}$), salinity tolerance (*Saltol*), and submergence tolerance (*Sub1*) in the genetic background of IWP. This

**Table 8. Performance of BILs and parents against submergence [29].**

| Entries | No. of seedlings/ Pot | No. of seedlings survived after submergence | Seedling Survival % | SES Score |
|---|---|---|---|---|
| IWP | 60 | 0 | 0 | 9 |
| FR13A | 60 | 52 | 86.7 | 5 |
| BIL#3-11-9-2 | 60 | 40 | 66.7 | 7 |
| BIL#3-11-11-1 | 60 | 48 | 80 | 5 |
| BIL#3-11-11-2 | 60 | 43 | 71.7 | 7 |

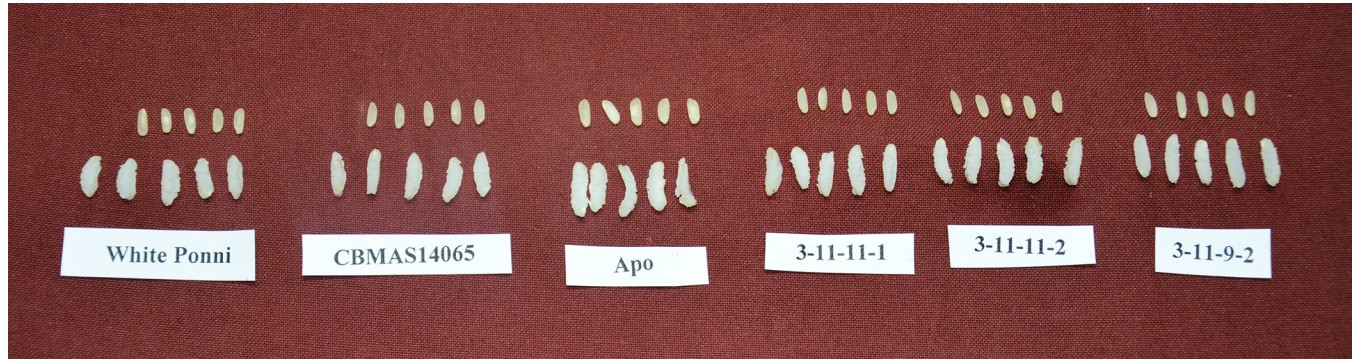

**Fig 10. Cooking quality characteristics of BILs, CBMAS14065 and IWP.**

paper describes the development of early generation back cross inbred lines of IWP pyramided with four different target QTLs and evaluation of early generation BILs for their target traits. In addition, the careful selection was imposed to select for semi-dwarf genotypes because IWP is prone to lodging because of its tall nature. $BC_1F_2$ progenies of IWP subjected to genotyping and progenies harboring homozygous donor alleles of $qDTY_{1.1}$, $qDTY_{2.1}$, *Saltol*, and *Sub1* were selected and their $BC_1F_3$ progenies were phenotyped for the target traits.

The success of any MAB program depends on identification of tightly linked markers for FGS and assessing recovery of recurrent parent genome using genome-wide polymorphic markers. In this study, a thorough survey among the parents in the target QTL regions resulted in the identification of four markers viz., RM472 ($qDTY_{1.1}$), RM2634 ($qDTY_{2.1}$), RM3412 (*Saltol*), and ART5 (*Sub1*) linked to the target QTLs, which were used for FGS. Use of molecular markers for BGS resulted in the identification of back cross progenies possessing the maximum recovery of the recurrent parent genome, thereby reducing the number of breeding cycles [47, 48]. All 3 BILs exhibited consistent performance with IWP, except for that of plant height. BILs registered a reduction of approximately 30 cm in height (94.2 to 100.7 cm) compared to that of IWP (133 cm), which enabled the development of a non-lodging version of

**Table 9. Grain and cooking quality traits of BILs.**

| Genotypes | KLBC (mm) | KBBC (mm) | KLBR | KLAC (mm) | KBAC (mm) | KLERC | Volume expansion ratio | Gelatinization Temp. | Gel consistency | Amylose % |
|---|---|---|---|---|---|---|---|---|---|---|
| IWP | 5.4 | 2 | 2.7 | 9.6 | 2.6 | 1.78 | 4.7 | Intermediate (69–74˚C) | Soft (66mm) | 24 (Intermediate) |
| CBMAS14065 | 5.2 | 1.8 | 2.9 | 9.6 | 2.6 | 1.85 | 4.1 | Intermediate (69–74˚C) | Soft (100mm) | 23 (Intermediate) |
| Apo | 5.8 | 2.4 | 2.4 | 10.8 | 3.4 | 1.86 | 4.8 | Intermediate (69–74˚C) | Medium (44mm) | 20 (Intermediate) |
| 3-11-11-1 | 5 | 1.8 | 2.8 | 9.6 | 2.6 | 1.92 | 4.9 | Intermediate (69–74˚C) | Soft (100mm) | 24 (Intermediate) |
| 3-11-11-2 | 5 | 1.7 | 2.9 | 9.2 | 2.6 | 1.84 | 4.8 | Intermediate (69–74˚C) | Soft (90mm) | 24.5 (Intermediate) |
| 3-11-9-2 | 5.1 | 1.8 | 2.8 | 9.6 | 2.8 | 1.88 | 4.8 | Intermediate (69–74˚C) | Soft (100mm) | 23 (Intermediate) |

KLBC, Kernel length before cooking; KBBC, kernel breadth before cooking; KLBR, length/breadth ratio; KLAC, kernel length after cooking; KBAC, kernel breadth after cooking, KLERC, kernel length elongation ratio on cooking

IWP. It could be attributed to the recombination between $qDTY_{1.1}$ and $sd1$ loci as reported earlier [49, 50].

BILs ($BC_1F_3$) of IWP (~ 90% of IWP genome) harboring $qDTY_{1.1}$, $qDTY_{2.1}$, $Saltol$, and $Sub1$ were evaluated for their tolerance against salinity, drought, and submergence stress to test the efficacy of these QTLs in combinations. Germination percent of IWP, FL478, and the BILs decreased with increasing NaCl concentration. At 80 mM NaCl stress, IWP exhibited only 66.7% germination, whereas FL478 (93.3%) and BILs (80–96.7%) recorded significantly higher germination. At 125 mM, IWP recorded only 26.7% germination, whereas all three BILs recorded more than 66.7%. This clearly indicated that introgression of $Saltol$ loci enabled enhanced tolerance against salinity as reported earlier [20, 21]. During vegetative stage stress, IWP recorded the maximum SES score of 9 (extreme susceptibility), whereas BILs exhibited an average SES score of 5. BILs recorded a lower Na/K ratio compared to that of IWP (Table 6). A lowered $Na^+/K^+$ ratio in the BILs may be attributed to their lower uptake of $Na^+$ primarily because of the effect of the $Saltol$ loci from FL478 in regulating the entry of Na salts [15, 51, 52].

The drought tolerance ability of plants can be evaluated either by visual scoring of leaf rolling, drying, and wilting symptoms reflecting the dehydration status of the plant tissues [53, 54]. In this study, BILs of IWP harboring $qDTY_{1.1}$ and $qDTY_{2.1}$ were evaluated for their ability to tolerate dehydration induced by polyethylene glycol. Hydroponically grown seedlings of BILs were exposed to PEG treatment along with IWP and scored for their responses to dehydration. IWP exhibited the score of 9 (highly susceptible), whereas BILs recorded a score of 5 (moderately tolerant). IWP exhibited a significant reduction (40%) in its root/shoot ratio during dehydration, whereas BILs showed only a 25% reduction in their root/shoot ratio. PEG-induced osmotic stress can cause hydrolysis of storage compounds, which lowers the internal osmotic potential [55]. Effect of $qDTY_{1.1}$ and $qDTY_{2.1}$ on protecting rice plants from dehydration-induced yield losses has already been reported by several authors [56, 57].

Evaluation of BILs of IWP harboring $Sub1$ loci from FR13A along with their parents revealed that all BILs exhibited increased recovery upon de-submergence after 12 days of flooding. The recurrent parent IWP exhibited complete susceptibility to 12 days of flooding, whereas the BILs showed 66–80% recovery (Table 8). This confirmed the effect of $Sub1$ in the genetic background of IWP as already demonstrated in various other genetic backgrounds, such as Swarna, Samba Mahsuri, TDK $Sub1$, BR 11 $Sub1$, CR 1009 $Sub1$, and CO43 $Sub1$ [16, 17, 24].

The grain quality characteristics of the BILs were found to be very similar to their parent IWP and CBMAS14065 (Table 9). Consumption of rice is determined by its grain length, shape, size and cooking quality. Generally, rice with more linear elongation gives finer appearance which is preferred by the consumers. The increased length elongation ratio in the grains of BILs after cooking was found to be on par with CBMAS14065 and IWP. Varieties with intermediate gelatinization temperature are preferred by the consumers and in this study, all three BILs were found to possess intermediate gelatinization temperature as that of their parents. Amylose content determines the softness and eating quality of cooked rice [58]. Grains of BILs were found to possess intermediate amylose content (23–24.5%) as that of their parents. Successful recovery of grain quality and cooking quality traits in the BILs with improved tolerance against salinity, drought and submergence is a significant achievement which will be readily accepted by the farmers.

During recent years, several attempts have been made to pyramid QTLs for stacking multiple traits in the genetic background of popular rice cultivars. Independent attempts have been made to develop rice lines tolerating various biotic stresses through MAS [59, 60]. Similarly, MAS has been successfully employed to develop abiotic stress tolerant genotypes viz.,

submergence-tolerant rice [17, 24], salinity-tolerant rice [20, 21, 61], and drought-tolerant rice [5, 62–64]. Limited attempts have been made to pyramid more than 1–2 QTLs controlling tolerance against major abiotic stresses, except for the development of a drought- and submergence-tolerant version of TDK1 [4] Attempts to combine tolerance against the three major abiotic stresses namely, drought, salinity, and submergence through MABB are limited. The present study clearly demonstrated that through a well-designed MAB strategy supported with limited phenotypic selection, yield and quality could be combined with tolerance against major abiotic stresses. Developed BILs could serve as valuable resources to further develop abiotic stress-tolerant rice varieties and also as genetic stock for the use in breeding programs.

## Supporting information

**S1 File. Raw_Images–Original uncropped raw images of gel pictures in the manuscript.** (PDF)

## Acknowledgments

Funding support of Department of Biotechnology, Government of India, New Delhi is acknowledged.

## Author Contributions

**Conceptualization:** Raveendran Muthurajan.

**Data curation:** Jagadeeshselvam Nallathambi, Hifzur Rahman, Rohit Kambale.

**Formal analysis:** Valarmathi Muthu, Ragavendran Abbai, Jagadeeshselvam Nallathambi, Hifzur Rahman, Bharathi Ayyenar.

**Investigation:** Valarmathi Muthu, Thiyagarajan Thulasinathan.

**Methodology:** Sasikala Ramasamy.

**Supervision:** Raveendran Muthurajan.

**Validation:** Hifzur Rahman, Rohit Kambale, Thiyagarajan Thulasinathan, Bharathi Ayyenar.

**Writing – original draft:** Sasikala Ramasamy, Raveendran Muthurajan.

**Writing – review & editing:** Raveendran Muthurajan.

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
