## [Decision Letter · Decision Letter 0]

10 Sep 2019

PONE-D-19-16885

Pyramiding QTLs controlling tolerance against drought, salinity and submergence in rice through marker assisted breeding

PLOS ONE

Dear Dr Muthurajan,

Thank you for submitting your manuscript to PLOS ONE. After careful consideration, we feel that it has merit but does not fully meet PLOS ONE’s publication criteria as it currently stands. Therefore, we invite you to submit a revised version of the manuscript that addresses the points raised during the review process.

This particularly relates to the agronomic characterization of the BCI lines.

We would appreciate receiving your revised manuscript by Oct 25 2019 11:59PM. To enhance the reproducibility of your results, we recommend that if applicable you deposit your laboratory protocols in protocols.io, where a protocol can be assigned its own identifier (DOI) such that it can be cited independently in the future. For instructions see: http://journals.plos.org/plosone/s/submission-guidelines#loc-laboratory-protocols

We look forward to receiving your revised manuscript.

Kind regards,

Paul C. Struik

Academic Editor

PLOS ONE

Journal Requirements:

Additional Editor Comments (if provided):

Reviewers' comments:

Reviewer's Responses to Questions

**Comments to the Author**

1. Is the manuscript technically sound, and do the data support the conclusions?

Reviewer #1: Yes

2. Has the statistical analysis been performed appropriately and rigorously? 

Reviewer #1: Yes

3. Have the authors made all data underlying the findings in their manuscript fully available?

Reviewer #1: No

4. Is the manuscript presented in an intelligible fashion and written in standard English?

Reviewer #1: No

5. Review Comments to the Author

Reviewer #1: Muthurajan and colleagues describe efforts to pyramid known genes for drought, salinity, and submergence tolerance into a single elite variety of rice with favored quality characteristics. The authors use a set of previously described microsat loci to characterize backcrosses, and then show that some of the backcross lines combine the stress tolerance of resistance donors with at least some of the quality traits of the elite parent. This is a nicely written breeding application. Although it does not break new ground theoretically, there are rather few examples of people doing this. This is a nice fit with PlosOne, as it is basically technically sound. It is the type of example that those teaching crop breeding or similar courses may utilize.

The authors have employed a set of microsats. These are effective in many breeding applications because they are cheap and easy to score. It would however be nice to have whole genome resequencing of the BCI lines to understand the scale of introgression from the resistance donor parents, to understand their footprint across the rice genome. This is not so much a criticism of this paper, but a suggestion for future work. Doing so would make this dataset much more novel. The interpretation would be much richer, and at this point resequencing rice genomes is a relatively trivial activity.

The authors have also not gone to great lengths to characterize the agronomic characteristics of the BCI lines. Although we are told they have the same cooking and quality characteristics as the elite parent, no data to this effect is shown. I have selected major revision on this ground. I believe the authors are likely to have this data, and that it should be included here in one manuscript. As with my point above, the extent of carryover of detrimental traits as part of pyramiding is a significant concern. Do these BCI lines really cook as well? Do they taste good? The taste of rice a key component of its cultural importance around the world. The bad appearance and taste of golden rice is its fatal flaw. This topic not only needs more space in the results, but should be mentioned in the introduction and elaborated on in the discussion. Simply stating the lines have good quality characteristics is not sufficient.

The manuscript is largely clearly written. I would encourage the authors to do another round of editing as a matter of good habits in the major revision phase. But I did not notice any clear typos, or issues with English grammatical usage such as tenses matching or proper article choice.

6. PLOS authors have the option to publish the peer review history of their article (what does this mean?). If published, this will include your full peer review and any attached files.

Reviewer #1: No

---

## [Author Response · Author response to Decision Letter 0]

10 Nov 2019

We have carried out precise experiments for assessing the grain and cooking quality characteristics of back cross progenies in comparison against the recurrent parent IWP and the results are included in the revised version.

---

## [Decision Letter · Decision Letter 1]

19 Dec 2019

Pyramiding QTLs controlling tolerance against drought, salinity and submergence in rice through marker assisted breeding

PONE-D-19-16885R1

Dear Dr. Muthurajan,

We are pleased to inform you that your manuscript has been judged scientifically suitable for publication and will be formally accepted for publication once it complies with all outstanding technical requirements.

With kind regards,

Paul C. Struik

Academic Editor

PLOS ONE

Additional Editor Comments (optional):

Reviewers' comments:

Reviewer's Responses to Questions

**Comments to the Author**

1. If the authors have adequately addressed your comments raised in a previous round of review and you feel that this manuscript is now acceptable for publication, you may indicate that here to bypass the “Comments to the Author” section, enter your conflict of interest statement in the “Confidential to Editor” section, and submit your "Accept" recommendation.

Reviewer #1: All comments have been addressed

2. Is the manuscript technically sound, and do the data support the conclusions?

Reviewer #1: Yes

3. Has the statistical analysis been performed appropriately and rigorously? 

Reviewer #1: Yes

4. Have the authors made all data underlying the findings in their manuscript fully available?

Reviewer #1: Yes

5. Is the manuscript presented in an intelligible fashion and written in standard English?

Reviewer #1: Yes

6. Review Comments to the Author

Reviewer #1: The authors have adequately addressed my comments. I believe those that teach crop breeding will find this a useful case study in the potential for molecular breeding. Several of the figures are at low resolution. For review this is fine, but for publication a higher resolution with greater image sharpness will be necessary.

7. PLOS authors have the option to publish the peer review history of their article (what does this mean?). If published, this will include your full peer review and any attached files.

Reviewer #1: No

---

## [Editor Report · Acceptance letter]

26 Dec 2019

PONE-D-19-16885R1 

Pyramiding QTLs controlling tolerance against drought, salinity, and submergence in rice through marker assisted breeding     

Dear Dr. Muthurajan:

I am pleased to inform you that your manuscript has been deemed suitable for publication in PLOS ONE. Congratulations! Your manuscript is now with our production department. 

With kind regards,

on behalf of

Prof. Paul C. Struik 

Academic Editor

PLOS ONE